# Comparison of the Eggshell and the Porcine Pericardium Membranes for Guided Tissue Regeneration Applications

**DOI:** 10.3390/biomedicines11092529

**Published:** 2023-09-13

**Authors:** Horia Opris, Mihaela Baciut, Marioara Moldovan, Stanca Cuc, Ioan Petean, Daiana Opris, Simion Bran, Florin Gligor Onisor, Gabriel Armenea, Grigore Baciut

**Affiliations:** 1Department of Maxillofacial Surgery and Implantology, Iuliu Hatieganu University of Medicine and Pharmacy, 400012 Cluj-Napoca, Romania; horia.opris@umfcluj.ro (H.O.); mbaciut@umfcluj.ro (M.B.); dr_brans@umfcluj.ro (S.B.); armencea.gabriel@umfcluj.ro (G.A.); gbaciut@umfcluj.ro (G.B.); 2Department of Polymer Composites, Institute of Chemistry Raluca Ripan, Babes-Bolyai University, 30 Fantanele Street, 400294 Cluj-Napoca, Romania; marioara.moldovan@ubbcluj.ro (M.M.); stanca.boboia@ubbcluj.ro (S.C.); 3Faculty of Chemistry and Chemical Engineering, Babes-Bolyai University, 11 Arany Janos Street, 400028 Cluj-Napoca, Romania; ioan.petean@ubbcluj.ro

**Keywords:** eggshell membrane, biocompatibility, bone regeneration, rat model

## Abstract

Guided bone regeneration is frequently used to reconstruct the alveolar bone to rehabilitate the mastication using dental implants. The purpose of this article is to research the properties of eggshell membrane (ESM) and its potential application in tissue engineering. The study focuses on the structural, mechanical, and histological characteristics of ESM extracted from *Gallus domesticus* eggs and to compare them to a commercially available porcine pericardium membrane (Jason^®^ membrane, botiss biomaterials GmbH, Zossen, Germany). Thus, histology was performed on the ESM, and a comparison of the microstructure through scanning electron microscopy and atomic force microscopy (AFM) was conducted. Also, mechanical tensile strength was evaluated. Samples of ESM were prepared and treated with alcohol for fixation and disinfection. Histological analysis revealed that the ESM architecture is constituted out of loose collagen fibers. However, due to the random arrangement of collagen fibers within the membrane, it might not be an effective barrier and occlusive barrier. Comparative analyses were performed between the ESM and the AFM examinations and demonstrated differences in the surface topography and mechanical properties between the two membranes. The ESM exhibited rougher surfaces and weaker mechanical cohesion attributed to its glycoprotein content. The study concludes that while the ESM displays favorable biocompatibility and resorb ability, its non-uniform collagen arrangement limits its suitability as a guided bone regeneration membrane in the current non-crosslinked native form. Crosslinking techniques may enhance its properties for such applications. Further research is needed to explore modifications and processing methods that could leverage the ESM’s unique properties for tissue engineering purposes.

## 1. Introduction

Alveolar bone resorption is a process that always takes place following tooth loss due to trauma or disease [1]. Thus, reconstruction has been proposed for a long time as a treatment to compensate for lost volume [2]. Furthermore, bone block and autologous bone graft has long been the golden standard for bone rebuilding [3]. In contrast, the concept of guided bone regeneration (GBR) requires compartmentalization, which employs a barrier membrane to permit the bone to heal [4]. In addition, Urban et al. [2] found that the membrane stabilizes the graft and the surrounding tissues.

Polytetrafluoroethylene (PTFE) and titanium meshes are the most common nonabsorbable materials used for GBR [5]. However, although they have excellent space maintenance, they require a second surgery to remove and carry a greater risk of exposure and contamination [6]. 

The alternative, resorbable membranes, may be of natural or synthetic origin. They can also be crosslinked or non-crosslinked, a process that modifies the membrane’s properties so that it absorbs water more slowly [7]. Consequently, they do not require surgical removal, but they are not suitable to maintain the space in large, particularly vertical, defects. As a matter of fact, exposure of the resorbable membranes does not always result in graft failure [8]. For instance, the bovine pericardium membranes have shown significant functional and morphological potential, which have provided the opportunity to examine cellular behavior [9].

Scanning electron microscopy (SEM) is used to evaluate the surfaces of biomaterials and bone [10]. Furthermore, it evaluates the tissue response to the graft materials and the production of the bone scaffold and osteogenic structures [11,12].

Atomic force microscopy (AFM) is used to examine the surface properties and the characteristics of materials [13]. In addition, it enables the evaluation of samples at the molecular level [14]. Moreover, by measuring forces and mapping the topography, AFM aids in evaluating the quality of graft materials, cellular adhesion, and proliferation and assists in studying the mechanical properties of regenerated bone [15]. 

In vivo studies on experimental animals’ subcutaneous tissue give valuable information on the histology-level effects a substance might produce [16]. Especially since the ideal membrane must have a surface that does not create tissue reaction, it is easy to manage, is semipermeable, and maintains a space for proper bone healing [17].

Although there are extensive inquiries regarding the use of the eggshell membrane (ESM) in clinical trials [18] and in animal use, much research is needed to fully evaluate the proprieties and potential. The eggshell membrane is of biologic origin, composed out of natural proteins, is a semipermeable membrane for oxygen and other nutrients, and acts as a barrier membrane for the contents of the egg [19]. These qualities and the research that is already in the public domain have produced questions regarding the actual biologic use for the membrane.

The aim of this paper is to assess the histologic and microscopic structure and tensile strength of the eggshell membrane in comparison to a commercially available porcine pericardium membrane (Jason^®^ membrane, botiss biomaterials GmbH, Zossen, Germany).

## 2. Materials and Methods

### 2.1. Production of the Membrane

Using the outer-shell membrane of *Gallus domesticus* eggshells, a collagen membrane was produced. The membrane was disinfected and fixed with a 99% alcohol solution.

The porcine pericardium membrane was procured from suppliers (Jason^®^ membrane, botiss biomaterials GmbH, Zossen, Germany) and was used for the testing described below.

Samples from each type of membrane (ESM and Jason) were produced for each test: for biocompatibility 1 × 1 cm (*n* = 12), samples for tensile testing (40 mm × 5 mm), for SEM (*n* = 12) and AFM (*n* = 12).

### 2.2. Histology

The specimens fixed with paraformaldehyde were mounted on glass slides, dehydrated with an alcohol gradient. Paraffin-embedded 10 mm sections were washed with xylene and rehydrated before being stained with hematoxylin and eosin (Histo-Lab. Ltd., Gothenburg, Sweden).

We adopted a specific histological processing technique with the goal of precisely preserving the structure for the assessment of the eggshell membrane. To achieve this, we cut a small pocket in the liver fragments of a recently sacrificed rat using a sharp scalpel, and then we inserted a piece of membrane into the pocket. The membrane and liver fragment were immersed in a 10% formalin solution for three days to fixate them. The fragments were cleared with 1-Butanol (Histo-Lab. Ltd., Gothenburg, Sweden), dehydrated with ethyl alcohol, and then embedded in paraffin when the fixing was finished. Hematoxylin and eosin (H&E) staining was used to create sections with a 5 μm thickness. An Olympus BX41 microscope (Olympus, Tokyo, Japan) with a digital image capture camera type E-330 was used to examine histological sections. The choice for this specific technique was because it enables us to produce a clear cross-section of a relatively soft material, but it can be hard to handle and process (the eggshell membrane). Also, it allows us not to damage the structure during handling and micro slicing.

### 2.3. Scanning Electron Microscopy

SEM investigation was executed with an Inspect S50 SEM Microscope produced by FEI Company, Hillsboro, OR, USA. The secondary electron images were obtained at an acceleration voltage of 25 kV at low vacuum mode without metallization.

### 2.4. Mechanical Testing: Tensile Testing

The mechanical strength of membranes was studied using a Lloyd LR5k Plus dual-column mechanical testing machine (Ametek/Lloyd Instruments, Meerbusch, Germany), equipped with a load cell with a maximum range of 5 kN. The tested samples had a rectangular shape measuring 40 mm high and 5 mm wide. The tensile test was performed with a separation of 25 mm at an expansion speed of 25 mm/min until they failed. The membranes were evaluated wet (immediately after removal from alcoholic solution) and dry (after 15 min of absorption on suction paper).

### 2.5. Atomic-Force Microscopy

The eggshell membrane was dried using a filter paper until it reached the proper conditions for the AFM investigation. The porcine pericardium membrane (Jason^®^ membrane) was extracted from the sterile envelope and a small corner was cut off for the AFM investigation. The investigation was setup with a JEOL JSPM 4210 Scanning Probe Microscope, produced by JEOL, Japan, Tokio. The probing cantilevers are of NSC 15 type produced by MicroMasch, Estonia, Talinn, with a resonant frequency of 325 kHz and a spring constant of 40 N/m. The topographic images were obtained at a scan rate of 1.5–3 Hz depending on the image size. The images were analyzed trough Jeol WIN SPM 2.0 processing soft and *Ra* (Surface roughness average) and *Rq* (root mean square roughness) roughness, measured for each image. At least three different macroscopic areas were scanned for each sample for a proper statistical average of the obtained values.

*Ra* represents the arithmetic average of the profile height and is described by Equation (1) and *Rq* root mean square of the profile height and is described by Equation (2):(1)$$Ra=\frac{1}{l_{r}}\int_{0}^{l_{r}}\left|z(x)\right|dx$$
and
(2)$$Rq=\sqrt{\frac{1}{l_{r}}\int_{0}^{l_{r}}\left|z{\left(x\right)}^{2}\right|dx}$$
where *l* is the profile length and *z* is the height at *x* point. Both *Ra* and *Rq* are important for a various research applications [20,21]. 

## 3. Results

### 3.1. Histopathologic Results

Histological fixation preserved the architecture of the eggshell membrane, which does not alter in contact with bodily fluids; fixation is similar to the process of cross-linking. After histological processing, the eggshell membrane remained intact, and it could be seen on the wide surface area in direct contact with the liver parenchyma (Figure 1 and Figure 2).

The eggshell membrane is structurally made up of collagen fibers arranged closely together but without a very rigorous organization. This connective tissue can be classified as dense, non-oriented connective tissue. Because non-oriented connective tissue does not have a very rigorous arrangement of collagen fibers, it can be easily appreciated that this membrane can be considered as a protective membrane but not as a separating membrane, as desired in guided bone regeneration. In other words, this membrane has good mechanical strength, but the random arrangement of collagen fibers means that meshes with a rigorous shape and size are not defined between them but are polymorphous in shape and size. In this regard, the membrane cannot be considered an efficient separating material as desired in guided bone regeneration because cells can pass through the larger meshes between collagen fibers, whereas the separating membrane should not allow this.

### 3.2. SEM Analysis

The eggshell membrane general aspect of the microstructure is presented in Figure 3a. The sample was positioned at 45° to the accelerated electron beam, revealing the membrane section on the middle horizontal position within the observation field, and the membrane surface is positioned below. Both exterior and interior sides of the membrane are visible, being strongly reticulated by collagen I type fibers, while the section cohesion is assured by only a few collagen type I fibers and a lot of collagen V type small fibers that bind the glycoprotein units. The eggshell membrane has a thickness of about 50 μm as observed in Figure 3a.

The microstructure of the eggshell membrane surface is presented in Figure 3b. The collagen I type network is clearly visible, having a lot of well interconnected fibers embedded into the glycoprotein matrix. Their average diameter is 2.5 ± 0.81 μm. The section microstructure, Figure 3c, reveals some vertical collagen type I fibers that interconnect both sides of the membrane, assuring its cohesion and consistency. Collagen type V fibers are smaller, having diameters of 0.6 ± 0.077 μm, and are predominantly horizontally oriented, assuring the texture base for the glycoprotein clusters from the membrane insight. The glycoprotein clusters are present, but it is exceedingly difficult to distinguish it from the collagen structure.

The porcine pericardium membrane is synthetically manufactured from collagen type I from porcine pericardium. Thus, it has a controlled interlaced structure based on the uniform collagen fibers. The overall microstructural aspect is presented in Figure 4a; both surface and section of the Jason membrane is visible due to the sample inclination of about 30° regarding the accelerated electron beam within SEM device. The surface is visible in the upper side of the observation field while the section is situated on the lower side of the SEM image in Figure 4a. The surface looks exceptionally smooth and is formed by a dense texture of collagen fibers. The section thickness has 100 ± 12 μm and is formed by several interlaced layers similar to the one observed in the surface. (There are about 10–12 successive collagen layers.)

Surface microstructure is presented in Figure 4b. It features a dense network of collagen fibers interlaced in a compact structure. It is difficult to observe each collagen fiber due to its small diameter, but some fascicles are clearly visible. Their general direction is situated from the lower left corner to the upper right corner of Figure 4b. The section microstructure has more visible details, Figure 4c. The successive layers are also interlaced together by a dense spatial texture of fine collagen fibers. The thickness of a single layer ranges from 1.5 to 2 μm, and the free space between two successive layers is situated at 10 ± 1.3 μm. Layer cross linking occurs through local intersections under a sharp angle (about 15–30°) situation; these are more visible in the left side of Figure 4c.

### 3.3. Atomic Force Microscopy Analysis

The eggshell membrane is a complex biological structure based on a very well reticulated collagen matrix based on both I and V types bonded with glycoprotein. Collagen type I is found in the outermost layers of the membrane while V type is characteristic for the deeper layers of the membrane. Thus, the topography of the eggshell membrane’s fine microstructure in Figure 5a reveals a dense network of collagen type I fibers with diameters of 2.5 ± 0.75 μm coated with a dense and compact glycoprotein layer that practically makes it impossible to visualize the tropocollagen units.

Figure 5b reveals a more detailed fine microstructure that evidences some small fibers of collagen V type in the range of 0.6–1 μm in diameter and are coated with a compact layer of rounded glycoprotein clusters of 300–400 nm. The nanostructural detail in Figure 5c reveals a single collagen V type fiber oriented from the upper left corner to the lower right corner with a diameter of 0.6 ± 0.08 μm, and glycoprotein clusters surrounds its structure.

Surface roughness, Figure 6, strongly depends on the topographic image scan side, the nanostructural aspects featuring lower roughness values (e.g., image side of 2.5 μm), while the fine microstructure has a slightly increased roughness due to the spatial orientation of the interlaced collagen fibers of type I and V. Thus, the roughness values of the fine microstructure ranges from *Ra* 173 to 194 nm and *Rq* 217 to 241 nm.

The Jason membrane is synthetically produced from collagen type I from porcine heart destined for dental application. The fine microstructure topography in Figure 7a reveals a dense texture of relative parallel fascicles of fine collagen fibers. Some of the fibers are interlaced under low angles assuring a good cohesion of the structure. It should enhance the tensile strength and avoid texture tearing under axial forces.

Figure 7b evidences a fine microstructure detail observed at a scan side of 2.5 μm, revealing the fibers interconnecting and assuring a good cohesion of the adjacent fascicles. Three concurrent fibers are situated in the lower left corner of Figure 7b and are bundled into an emergent one that passes through the right side of the structure being incorporated in the next fascicle. The nanostructural detail in Figure 7c allows us to properly observe and measure the fiber’s diameter that is situated at 160 ± 15 nm. The tropocollagen units are clearly visible in the fiber’s structure as rounded elements with a diameter of 67 nm.

The Jason membrane proves to be flatter than the eggshell membrane due to the synthetic interlacing and due to the lack of the glycoprotein matrix. Thus, the roughness values are less dependent on the topographic image side. However, at the nanostructural level, the roughness is slightly lower, Figure 8. The fine microstructure presents roughness ranges as follows: *Ra* 42.6 to 49.4 nm and *Rq* 54.6 to 63.5 nm.

### 3.4. Mechanical Proprieties

The results of the mechanical testing of the membranes can be observed in Table 1, and the descriptive statistics are in Table 2. The mean of the dried membranes is 10.42 for the eggshell membrane and, for the Jason, 65.72. The same difference can be seen for the membranes observed that were immersed in SBF (simulated body fluids). This is in accordance with the histologic results and the SEM and AFM analysis. The lack of structure and the nonuniform distribution of the collagen fibers determine such a significant difference between the membranes results of the mechanical testing.

In Figure 9, the tensile strength curve for dry eggshell membrane after 15 min (upper left graph), dry eggshell membrane immersed in SBF (upper right graph), Jason membrane (lower left graph), and Jason membrane immersed in SBF (lower right graph) can be compared. The eggshell membrane has a higher load and break for the dried eggshell membrane. The membrane itself has significantly lower values for the load at break. The porcine pericardium origin of the collagen membrane offers a denser structure with a thicker layer which, in fact, gives it better handling capabilities and resistance at tensile testing.

## 4. Discussion

Due to the content of the membrane, it is biocompatible and resorbable. Despite this, the shell membrane cannot be an ideal separating membrane in guided bone repair because the collagen fibers are not rigorously arranged, so they do not form regular meshes of a suitable and comparable size, which could prevent cell migration through the membrane.

An early study of biocompatibility by Rothamel et al. [22] regarding the porcine pericardium membranes has revealed that they display a fibrous structure, cell proliferation, and no inflammatory reaction. Moreover, the degradation of the membranes has been shown to be around 4 to 8 weeks—for some commercial variants, even 12 weeks.

Nevertheless, the rationale behind bone regeneration techniques, not dependent on the technique itself, involves the facilitation of three-dimensional dental implant placement in a correct prosthetic position for functional loading [23]. Hence, the histologic changes that occur during alveolar bone healing can be summed up into subsequent processes: inflammatory, proliferative, and remodeling. First, it involves the blood clot formation and the inflammatory cells’ migration. Secondly, it includes fibroplasia and woven bone formation. Then, the remodeling stage will reshape the alveolar bone considering the architecture and the functional loading. In conclusion, after tooth loss, it is expected that at least 50% bone loss after healing will occur, especially on the buccal side and more volume in the molar region [24]. Furthermore, the alveolar bone remodeling can be influenced by using xenograft in combination with a collagen membrane; thus, the resorption rate can be decreased. In addition, the facial aspect of the bone always resorbs in a small portion [25]. Above all, bone remodeling occurs, and usually the alveolar bone stabilizes at 9 months postoperative after GBR, to a significant degree, although it is still a better outcome than no treatment [26]. 

Although both resorbable and non-resorbable membranes produce the same quality of bone, when using titanium meshes, it was observed that the thickness and the stability after 1 year of loading was greater [27]. Despite all the efforts, the hard tissues resulting from GBR using resorbable membranes with xenograft material will always retract during the remodeling phase. Furthermore, Jiang et al. [28] concluded that new bone formation further than the bony envelope is not predictable.

What is a more important issue to discuss is the quality of studies found in the literature regarding the follow-up of clinical cases. In general, the median follow-up is about 60 months, not enough for what one could consider a success, and there is also a lack of prospective studies which compare different techniques and assess the long-term stability and outcome [23]. 

In fact, Bornert et al. [29] found that the barrier function of the pericardium porcine membrane is similar to other commercially available collagen membranes (Biogide) in regard to the resorption kinetics and resorption rate. Also, the barrier function was intact at 12 weeks, and the cell adhesion facilitated the bone formation process.

Similarly, a study on a critically sized rat calvaria defect by You et al. [30] found that the pericardium membrane, which was compared to native collagen membrane at 4 weeks, showed good biocompatibility, good barrier function, and enhanced bone regeneration. Furthermore, its surface promotes the proliferation and differentiation of the human bone mesenchymal stem cells, increased alkaline phosphatase activity, and upregulated expression of bone-specific genes.

In their in vivo tissue response study, Radenković et al. [31] compared a commercially available cross-linked porcine sugar membrane with two non-cross-linked collagen membranes. They concluded that all membranes lead to similar bone formation, but the cross-linked membrane is more stable and resorbs more slowly up to 60 days after implantation. In research recently publicized regarding an in vivo model which compares a novel membrane based on bovine dermis collagen with two commercially available alternatives, the authors found no difference between the proposed product regarding its biocompatibility [32]. Comparably, another study on calvaria’s critically sized defects revealed that collagen porcine membranes promote new bone formation when compared to a negative control [33]. 

Strnková et al. [34] evaluated the tensile behavior of different eggshell membranes (hen, goose, and Japanese quail) and revealed that they express linear and non-linear tensile deformation. Also, the parameters increased with the loading rate, with the smallest values being measured for the quail eggs and the highest for the goose eggs. Moreover, the structure relationship of the eggshell membrane has been proven that it behaves both as Mooney–Rivlin and Hookean materials in different environmental conditions. As a result, it can stretch and restore its position, or it can have a nonlinear behavior akin to rubber [35,36]. 

Because of the AFM investigation, both membranes revealed significant structural and topographical differences. Nevertheless, the eggshell membrane is rougher than that of the porcine pericardium membrane due to the interpenetration of collagen types I and V bonded together with rounded glycoprotein structure while Jason membrane is synthetically textured by size-controlled collagen type I fibers of about 160 nm diameter.

Also, the microstructural aspect indicates that porcine pericardium membrane has a constant mechanical behavior under tensile strength testing due to the very well interlaced collagen fascicles. Another point is that the eggshell membrane cohesion strongly depends on the glycoprotein structure, assuring collagen network binding, whereas, in a wet state, it might exhibit significant tensile strength which might strongly decrease after glycoprotein drying because of cohesion becoming fragile.

In particular, tensile strength membranes can be influenced by a number of factors: origin of the material, processing methods, and crosslinking. Significantly, glutaraldehyde treatment of the membrane has been shown to enhance its proprieties, reduce the resorption time, and increase tensile strength [37]. Raz et al. [38], in a mechanical testing paper, examined three membranes and found comparable mechanical results as in our study: the dry state exhibited higher tensile strength values, and the denser the membrane the higher the output values.

Overall, the limitation factor for this study is the fact that it lacks a dynamic histologic and histomorphometry image of the soft and hard tissue reaction to the membrane for comparison. Also, a radiologic image could be of help to evaluate the potential of bone regeneration.

Above all, future research is needed to completely assess the membrane in an in vivo model with a bone defect. Also, investigation is needed regarding the crosslinking of the membrane, a process which might improve the mechanical and barrier functions. In addition, there could be a future perspective to research and perform an ultrastructural analysis using transmission electron microscopy.

## 5. Conclusions

The membrane does not determine a foreign body reaction. It can be used as an occlusive barrier but not as a separation membrane. It is a potential vehicle for other substances that may enhance its proprieties. Because it consists of collagen, it is highly biocompatible, resorbable, and biodegradable by the organism, and it does not induce a foreign body response.

Crosslinking may enhance the eggshell membranes proprieties to become a feasible, guided bone regeneration product. Further research is needed to better understand the processing needed to utilize this cheap and readily available biomaterial.

## Figures and Tables

**Figure 1 biomedicines-11-02529-f001:**
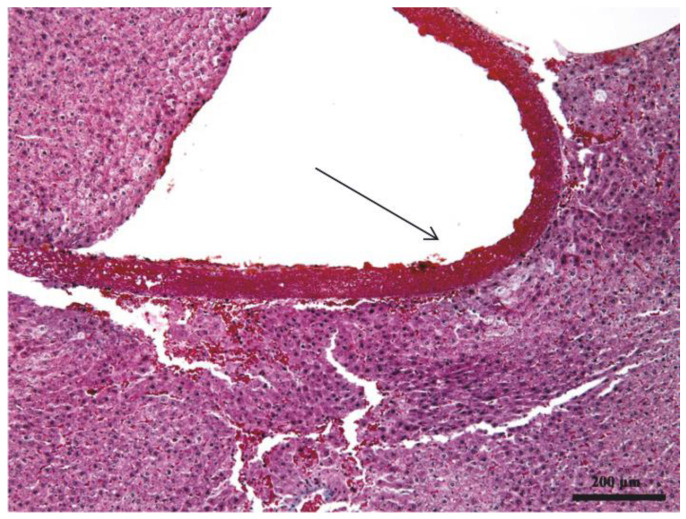
Intact eggshell membrane (black arrow) preserved by histological fixation, visible in direct contact with liver parenchyma (Hematoxylin eosin staining, 10×).

**Figure 2 biomedicines-11-02529-f002:**
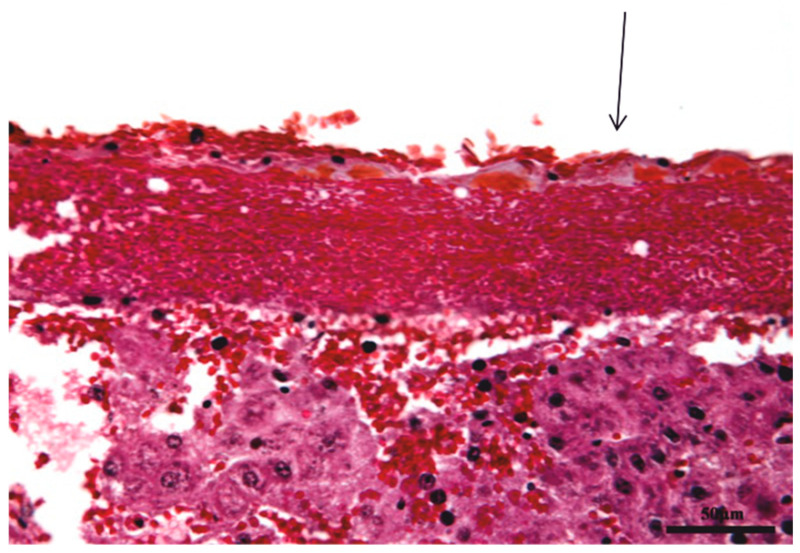
Eggshell membrane (black arrow) architecture remains unchanged due to histological fixation, comparable to cross-linking process; collagen fibers packed together can be seen without a rigorous union (Hematoxylin eosin staining, 40×).

**Figure 3 biomedicines-11-02529-f003:**
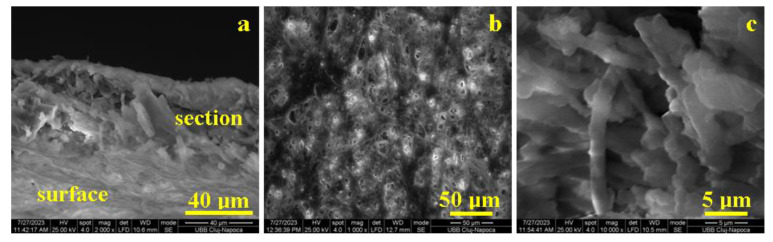
SEM images for the eggshell membrane: (**a**) ensemble microstructural view, (**b**) surface microstructure, and (**c**) section microstructure.

**Figure 4 biomedicines-11-02529-f004:**
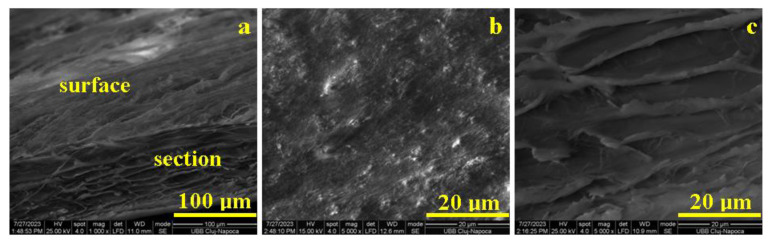
SEM images for Jason membrane: (**a**) ensemble microstructural view; (**b**) surface microstructure, and (**c**) section microstructure.

**Figure 5 biomedicines-11-02529-f005:**
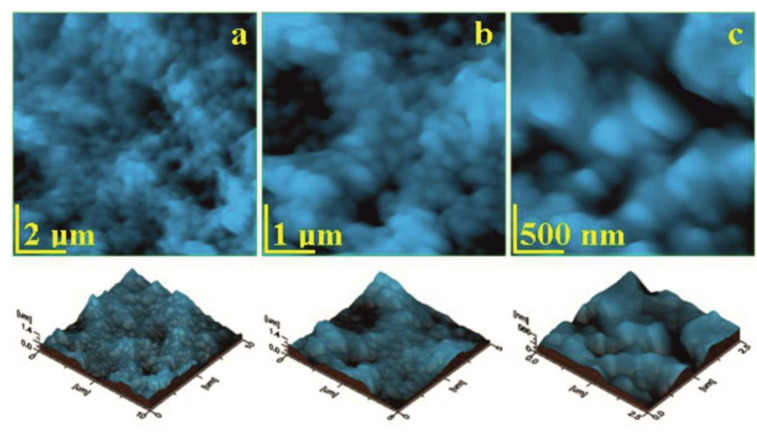
Eggshell membrane topographic images scanned at different images sides: (**a**) 10 μm, (**b**) 5 μm, and (**c**) 2.5 μm.

**Figure 6 biomedicines-11-02529-f006:**
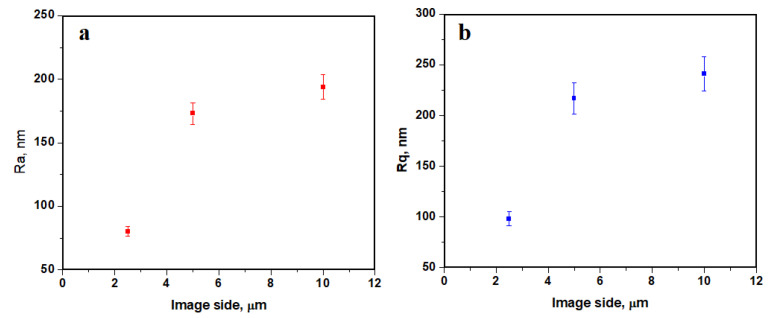
Roughness variation with the image side: (**a**) *Ra* and (**b**) *Rq*.

**Figure 7 biomedicines-11-02529-f007:**
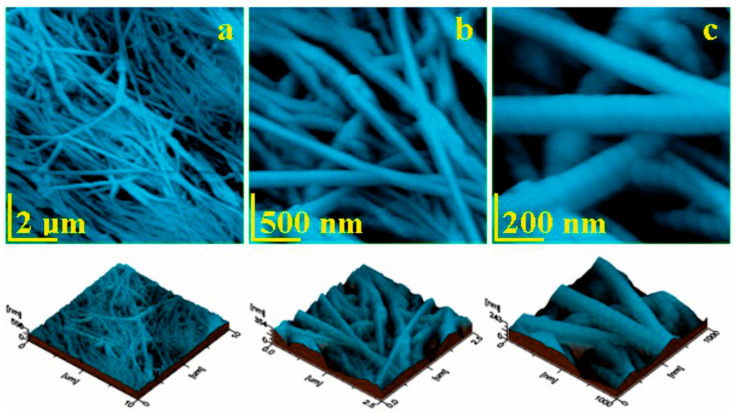
Jason membrane topographic images scanned at different images sides: (**a**) 10 μm, (**b**) 2.5 μm, and (**c**) 1 μm.

**Figure 8 biomedicines-11-02529-f008:**
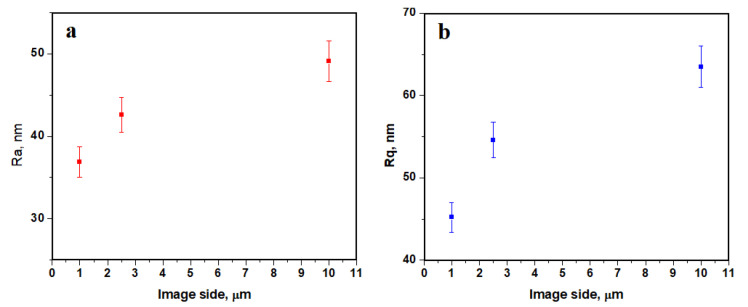
Roughness variation with the image side: (**a**) *Ra* and (**b**) *Rq*.

**Figure 9 biomedicines-11-02529-f009:**
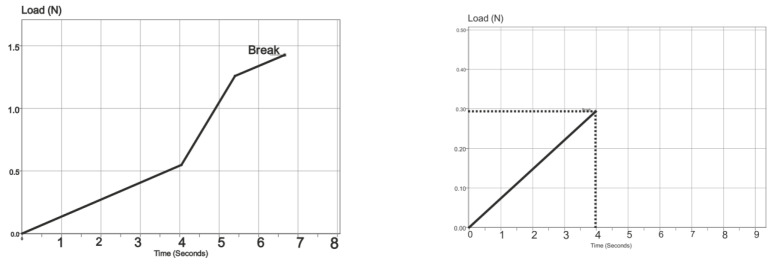
Tensile strength curve for dry eggshell membrane after 15 min (upper left graph), dry eggshell membrane immersed in SBF (upper right graph), Jason membrane (lower left graph), Jason membrane immersed in SBF (lower right graph).

**Table 1 biomedicines-11-02529-t001:** Tensile Strength (MPa) testing on the eggshell membrane comparing to the Jason membrane.

Eggshell Membrane after 15 min of Drying	Dried Jason Membrane	Eggshell Membrane Immersed in SBF and Dried 15 min	Jason Membrane Immersed in SBF and Dried 15 min
4.93	59.65	1.20	61.91
13.74	67.01	1.20	64.22
10.09	62.25	1.27	64.32
8.35	67.01	1.25	69.63
4.36	62.25	1.25	69.87
6.37	67.01	1.23	74.44
10.75	66.08	1.22	69.73
11.70	65.78	1.24	67.00
12.46	65.65	1.23	74.18
11.82	59.91	1.21	74.14

**Table 2 biomedicines-11-02529-t002:** Descriptive statistics of the tensile strength (MPa) of the eggshell membrane and Jason membrane.

	Eggshell Membrane after 15 min of Drying	Dried Jason Membrane	Eggshell Membrane Immersed in SBF and Dried 15 min	Jason Membrane Immersed in SBF and Dried 15 min
Mean	9.46	64.26	1.23	68.94
Median	10.42	65.72	1.23	69.68
Standard Deviation	3.29	2.95	0.02	4.52
Minimum	4.36	59.65	1.20	61.91
Maximum	13.74	67.01	1.27	74.44

## Data Availability

Data sharing not applicable.

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
