# Peer review of "Comparison of the Eggshell and the Porcine Pericardium Membranes for Guided Tissue Regeneration Applications"

_biomedicines, 2023, doi:10.3390/biomedicines11092529_

Round 1

Reviewer 1 Report

The aim of this manuscript is to investigate the properties of eggshell membrane and its potential application in tissue engineering.

This manuscript shows rich content, providing a deep insight for some works: the study is within the journal’s scope, and I found it to be well-written, providing sufficient information. Even if the manuscript provides an organic overview, with a densely organized structure and based on well-synthetized evidence, there are some suggestions necessary to make the article complete and fully readable. For these reasons, the manuscript requires major changes.

Please find below an enumerated list of comments on my review of the manuscript:

INTRODUCTION:

LINE 43: Guided bone regeneration (GBR).

LINE 43: GBR improves and guides the mechanism of bone repair, by requiring scaffold materials with a three-dimensional structure, similar to the extracellular matrix (see, for reference: http://dx.doi.org/10.1016/b978-0-12-815732-9.00068-1).

LINE 53: Bioresorbable membranes, made of collagen derived from human, porcine or bovine pericardium, are frequently employed in clinical routine. Specifically, the bovine pericardium membranes showed significant functional and morphological potential, which provides the opportunity to examine cellular behavior, also in different fields. Furthermore, the bovine pericardium membranes are decellularized to remove antigenic epitopes associated with cell membranes and intracellular components, thus to enhance the biocompatibility to the wound site (see, for reference: https://doi.org/10.3390/ma15238284). This is the major concern of this manuscript: this manuscript may benefit from providing an organic and up-to-date description of the pivotal role, played by bioresorbable membranes, in regenerative procedures.

Besides, the authors should provide a list of the abbreviations, mentioned in this manuscript.

RESULTS:

LINE 142-147: The authors should express the magnification and scalebar of the LM data.

CONCLUSIONS:

LINE 369: In this conclusive section, the authors should mention, among the future perspectives of this study, the possibility to perform also an ultrastructural analysis, by TEM.

The main topic is interesting, and certainly of great clinical impact. As regards the originality and strengths of this manuscript, this is a significant contribute to the ongoing research on this topic, as it extends the research field on the properties of eggshell membrane and its potential application in tissue engineering. Overall, the contents are rich, and the authors also give their deep insight for some works.

As regards the section of methods, there is a specific and detailed explanation for the methods used in this study: this is particularly significant, since the manuscript relies on a multitude of methodological and statistical analysis, to derive its conclusions. The methodology applied is overall correct, the results are reliable and adequately discussed.

The conclusion of this manuscript is perfectly in line with the main purpose of the paper: the authors have designed and conducted the study properly. As regards the conclusions, they are well written and present an adequate balance between the description of previous findings and the results presented by the authors.

Finally, this manuscript also shows a basic structure, properly divided and looks like very informative on this topic. Furthermore, figures and tables are complete, organized in an organic manner and easy to read.

In conclusion, this manuscript is densely presented and well organized, based on well-synthetized evidence. The authors were lucid in their style of writing, making it easy to read and understand the message, portrayed in the manuscript. Besides, the methodology design was appropriately implemented within the study. However, many of the topics are very concisely covered. This manuscript provided a comprehensive analysis of current knowledge in this field. Moreover, this research has futuristic importance and could be potential for future research. However, major concerns of this manuscript are with the introductive section, the results and conclusive section: for these reasons, I have major comments for these sections, for improvement before acceptance for publication. The article is accurate and provides relevant information on the topic and I have some major points to make, that may help to improve the quality of the current manuscript and maximize its scientific impact. I would accept this manuscript if the comments are addressed properly.

Minor editing of English Language are required.

Author Response

Reviewer 1:

The aim of this manuscript is to investigate the properties of eggshell membrane and its potential application in tissue engineering.

This manuscript shows rich content, providing a deep insight for some works: the study is within the journal’s scope, and I found it to be well-written, providing sufficient information. Even if the manuscript provides an organic overview, with a densely organized structure and based on well-synthetized evidence, there are some suggestions necessary to make the article complete and fully readable. For these reasons, the manuscript requires major changes.

Please find below an enumerated list of comments on my review of the manuscript:

Point 1:

INTRODUCTION:

LINE 43: Guided bone regeneration (GBR).

Response 1: Thank you very much for the observation. We have added the necessary text.

Point 2:

LINE 43: GBR improves and guides the mechanism of bone repair, by requiring scaffold materials with a three-dimensional structure, similar to the extracellular matrix (see, for reference: http://dx.doi.org/10.1016/b978-0-12-815732-9.00068-1).

Response 2:

Thank you so much for the insight. We have modified the paragraph accordingly.

Point 3:

LINE 53: Bioresorbable membranes, made of collagen derived from human, porcine or bovine pericardium, are frequently employed in clinical routine. Specifically, the bovine pericardium membranes showed significant functional and morphological potential, which provides the opportunity to examine cellular behavior, also in different fields. Furthermore, the bovine pericardium membranes are decellularized to remove antigenic epitopes associated with cell membranes and intracellular components, thus to enhance the biocompatibility to the wound site (see, for reference: https://doi.org/10.3390/ma15238284). This is the major concern of this manuscript: this manuscript may benefit from providing an organic and up-to-date description of the pivotal role, played by bioresorbable membranes, in regenerative procedures.

Besides, the authors should provide a list of the abbreviations, mentioned in this manuscript.

Response 3:

Thank you so much! We have added a paragraph and have corrected the abbreviations according to the instructions for authors from Biomedicines/MDPI.

Point 4:

RESULTS:

LINE 142-147: The authors should express the magnification and scalebar of the LM data.

Response:

Thank you very much. We have added magnification and scalebar.

Point 5:

CONCLUSIONS:

LINE 369: In this conclusive section, the authors should mention, among the future perspectives of this study, the possibility to perform also an ultrastructural analysis, by TEM.

Response 5:

Thank you very much for your suggestion. Will certainly plan for this soon!

Point 6:

The main topic is interesting, and certainly of great clinical impact. As regards the originality and strengths of this manuscript, this is a significant contribute to the ongoing research on this topic, as it extends the research field on the properties of eggshell membrane and its potential application in tissue engineering. Overall, the contents are rich, and the authors also give their deep insight for some works.

As regards the section of methods, there is a specific and detailed explanation for the methods used in this study: this is particularly significant, since the manuscript relies on a multitude of methodological and statistical analysis, to derive its conclusions. The methodology applied is overall correct, the results are reliable and adequately discussed.

The conclusion of this manuscript is perfectly in line with the main purpose of the paper: the authors have designed and conducted the study properly. As regards the conclusions, they are well written and present an adequate balance between the description of previous findings and the results presented by the authors.

Finally, this manuscript also shows a basic structure, properly divided and looks like very informative on this topic. Furthermore, figures and tables are complete, organized in an organic manner and easy to read.

In conclusion, this manuscript is densely presented and well organized, based on well-synthetized evidence. The authors were lucid in their style of writing, making it easy to read and understand the message, portrayed in the manuscript. Besides, the methodology design was appropriately implemented within the study. However, many of the topics are very concisely covered. This manuscript provided a comprehensive analysis of current knowledge in this field. Moreover, this research has futuristic importance and could be potential for future research. However, major concerns of this manuscript are with the introductive section, the results and conclusive section: for these reasons, I have major comments for these sections, for improvement before acceptance for publication. The article is accurate and provides relevant information on the topic and I have some major points to make, that may help to improve the quality of the current manuscript and maximize its scientific impact. I would accept this manuscript if the comments are addressed properly.

Response 6:

Thank you very much for your effort and your effort to make our research better! We kindly appreciate the comments and suggestions put together and hopefully have answered them to fulfill your expectations.

Reviewer 2 Report

In the present 13-page article, "Comparison of the eggshell and the porcine pericardium membranes for guided tissue regeneration applications", the authors describe eggshell membrane as a biomaterial for tissue regeneration and assasses the histologic, microscopic structure and tensile strength of the eggshell membrane in comparison to a commercially available porcine pericardium membrane. This is an exciting approach, but some questions remain unanswered and extensive changes to the discussion are necessary. The following are my comments and remarks:

 Line 44: The introduction should be revised as it does not read well at the moment: Sentence [citation] Sentence [citation]. Variations such as "Guided bone regeneration requires compartmentalization, which uses a barrier membrane to allow the bone to heal.  In addition, Urban et al [2] found that the membrane stabilized the graft and surrounding tissue." would be more varied for the reader. 

Line 47: GBR abbreviation not explained

Line 51: I would also compact the text, i.e. remove the blanks in line: 51, 56, 61, 65 and 68.

Line 80: ESM not explanined

Line 81: how many samples were used for SEM and AFM?

Line 84: origin of the chemicals used, not given

Line 104: were the tensile measurings performed according to a specific DIN EN ISO / ASTM ? How long was measured (1 minute; 30 seconds)?

Line 108: was the tensile test carried out until the specimens failed?

Line 113: why didn’t you freeze dry the membranes (then it stays in “shape” and you can see the “swollen” state)

Line 141/Figure 1: I would mark the ESM with arrows in the image; scalebar is missing

Line 144/Figure 2: see above;

Line 172: Can this be seen at all in the present magnification? I would consider showing this also in the magnification of Fig. 3c; especially when the authors write in line 175 that the fibers have a diameter of 0.6 µm.

Line 184: "...SEM Image in Figure 2a..." (must be Figure 4a I guess)

Line 191-201: Can you be sure that the surface structures found are not drying artifacts or negative imprints of the paper used for drying?

Line 192: why are the magnifications in fig. 3 and 4 different (40, 50, 5 µm vs 100, 20, 20 µm)? For comparability, both images should have identical magnifications. Furthermore, as the author, I would consider whether it would be better to combine the two figures for better comparability.

Line 208: “about 2.5 µm” is not scientific – please specify the exact values (e.g. 2.4 ± 0.4 µm)

Line 224: Does it makes sense to show Rq in addition to Ra and not better Rz, especially because Ra and Rq are mean values and Rz rather shows the actual conditions?

Line 231: I thought the jason fibres were supposed to be porcine pericardium, to me it looks more like electrospun nonwovens

Line 231ff: in the freeze-dried "swollen" state, the difference between the membranes would probably not be so great, except for the missing round glycoprotein clusters at the ESM

Line 232 Fig5/7: the same as for fig 3/4 also applies to fig 5/7: why are the magnifications of the AFM images different? In terms of comparability, it would make sense to show the same magnifications here as well. 

Line 239: “around 160 nm” is not scientific – please specify the exact values (e.g. 148 ± 17 nm)

Line 241: “about 67 nm” see above

Line 251, 252, Table 1, 2: please used (.) instead of (,)

Line 251/252: unit [MPa] is missing in the text

Line 260: must be MPa instead of Mpa

Line 270/Figure 9: Please revise the images, the mesh in the background is not different from the trace, please draw thicker or highlight in color, also the font is not readable

Line 277/Figure 10: on the pictures in Fig. 10 it looks as if the clamping and pulling was not straight - if this is the case, all measured values of the tensile test are wrong - clamping must always be straight in all 3 axes

Line 279 ff: up to line 360, the discussion reads like an introduction; in the discussion, your own values are to be classified in relation to known values from the literature. In addition, a critical consideration of the literature or one's own measured values can be made. In the present form, the discussion needs to be revised considerably. 

Author Response

Reviewer 2

In the present 13-page article, "Comparison of the eggshell and the porcine pericardium membranes for guided tissue regeneration applications", the authors describe eggshell membrane as a biomaterial for tissue regeneration and assesses the histologic, microscopic structure and tensile strength of the eggshell membrane in comparison to a commercially available porcine pericardium membrane. This is an exciting approach, but some questions remain unanswered and extensive changes to the discussion are necessary. The following are my comments and remarks:

Point 1:

 Line 44: The introduction should be revised as it does not read well at the moment: Sentence [citation] Sentence [citation]. Variations such as "Guided bone regeneration requires compartmentalization, which uses a barrier membrane to allow the bone to heal.  In addition, Urban et al [2] found that the membrane stabilized the graft and surrounding tissue." would be more varied for the reader. 

Response 1:

Thank you for your remark. We agree that the readability must be improved. Significant effort has been put into rephrasing the introduction to make it easy for the reader.

Point 2:

Line 47: GBR abbreviation not explained

Response 2: We have added the abbreviation. Thank you!

Point 3:

Line 51: I would also compact the text, i.e. remove the blanks in line: 51, 56, 61, 65 and 68.

Response 3: Thank you very much. We have removed the unnecessary blanks.

Point 4:

Line 80: ESM not explanined.

Response 4: We have added the explanation for the ESM.

Point 5:

Line 81: how many samples were used for SEM and AFM?

Response 5: Three similar samples were probed by SEM and AFM. The relevance of the SEM and AFM images was assured by the investigation of at least 3 different macroscopic areas on each sample. The average results were presented.

Point 6:

Line 84: origin of the chemicals used, not given

Response 6: We have added the necessary details.

Point 7:

Line 104: were the tensile measurings performed according to a specific DIN EN ISO / ASTM ? How long was measured (1 minute; 30 seconds)?

Response 7: The measurements were carried out until the samples failed. No the tests were not according to any ISO standard.

Point 8:

Line 108: was the tensile test carried out until the specimens failed?

Response 8: Yes until the samples failed.

Point 9:

Line 113: why didn’t you freeze dry the membranes (then it stays in “shape” and you can see the “swollen” state)

Response 9: To our best effort, we and our pathologist have not yet managed to properly histologically assess the membrane properly by freezing it. We believe that this method is as close as possible to the unaltered state.

Point 10:

Line 141/Figure 1: I would mark the ESM with arrows in the image; scalebar is missing

Response 10: Thank you. We have added the necessary details.

Point 11:

Line 144/Figure 2: see above;

Response 11: Thank you. We have added the necessary details.

Point 12:

Line 172: Can this be seen at all in the present magnification? I would consider showing this also in the magnification of Fig. 3c; especially when the authors write in line 175 that the fibers have a diameter of 0.6 µm.

Response 12: Eggshell membrane and Jason membrane have different morphology with details at different sizes, they have quite different morphology. Therefore, the magnifications were chosen in order to reveal better the morphological aspects of the samples. Thus, the general aspect was presented in figures 3a and 4a; surface aspect in figures 3b and 4b and section morphology in figures 3c and 4c. Each image was taken at an optimal magnification for revealing the microstructural aspects.

Point 13:Line 184: "...SEM Image in Figure 2a..." (must be Figure 4a I guess)

Response 13: Yes, it is 4a. We modified it.

Point 14: Line 191-201: Can you be sure that the surface structures found are not drying artifacts or negative imprints of the paper used for drying?

Response 14: The SEM samples were not dried using paper towels, but under the vacuum of the SEM microscope.

Point 15: Line 192: why are the magnifications in fig. 3 and 4 different (40, 50, 5 µm vs 100, 20, 20 µm)? For comparability, both images should have identical magnifications. Furthermore, as the author, I would consider whether it would be better to combine the two figures for better comparability.

Response 15: Eggshell membrane and Jason membrane have different morphology with details at different sizes, they have quite different morphology. Therefore, the magnifications were chosen in order to reveal better the morphological aspects of the samples. Thus, the general aspect was presented in figures 3a and 4a; surface aspect in figures 3b and 4b and section morphology in figures 3c and 4c. Each image was taken at an optimal magnification for revealing the microstructural aspects.

Point 16: Line 208: “about 2.5 µm” is not scientific – please specify the exact values (e.g. 2.4 ± 0.4 µm)

Response 16: Thank you very much. We modified accordingly.

Point 17: Line 224: Does it makes sense to show Rq in addition to Ra and not better Rz, especially because Ra and Rq are mean values and Rz rather shows the actual conditions?

Response 17: Thank you for the comment. We measure the roughness of the whole scanned surface following Ra and Rq parameters. These measurements are sustained by the tri-dimensional profiles presented below of each topographic image. There were investigated at least 3 different macroscopic areas to assure a proper statistical relevance and the mean values were presented in Figures 6 and 8.

Rz represent the maximum peak to valley height of the profile, within a single sampling length. On the other words, Rz describes only some particular aspects regarding a single bi-dimensional profile which might be affected by some particular local characteristics which might not be relevant for the whole surface. 

Point 18: Line 231: I thought the jason fibres were supposed to be porcine pericardium, to me it looks more like electrospun nonwovens

Response 18: Response 18: Indeed the morphological aspect is similar to the electrospuns but has no connection to the electrospining process. Jason membrane is made by collagen extracted from porcine pericardium but the fibers were not used as obtained. They were textured during the woven process within manufacturing.

Point 19: Line 231ff: in the freeze-dried "swollen" state, the difference between the membranes would probably not be so great, except for the missing round glycoprotein clusters at the ESM

Point 20: Line 232 Fig5/7: the same as for fig 3/4 also applies to fig 5/7: why are the magnifications of the AFM images different? In terms of comparability, it would make sense to show the same magnifications here as well. 

Response 20: The magnifications were chosen in order to reveal better the morphological and topographic aspects of each investigated material: from the fine microstructure to the nanostructural details.

Images comparability is assured by AFM images taken at the same scanned area such as:

- Figure 5a to Figure 7a at the image side of 10 µm for the fine microstructure

- Figure 5c to Figure 7b at the image side of 2.5 µm for the nanostructural aspects.

The eggshell membrane required an intermediary magnification Figure 5b observed at an image side of 5 µm.

The Jason membrane required a nanostructural detail presented in Figure 7c observed at an image side of 1 µm.

Point 21: Line 239: “around 160 nm” is not scientific – please specify the exact values (e.g. 148 ± 17 nm)

Response 21: Thank you, it was revised as 160 ± 15 nm.

Point 22: Line 241: “about 67 nm” see above

Response 22: Thank you, it was revised

Point 23: Line 251, 252, Table 1, 2: please used (.) instead of (,)

Response 25: We modified it.

Point 24: Line 251/252: unit [MPa] is missing in the text

Response 25: We modified it.

Point 25: Line 260: must be MPa instead of Mpa

Response 25: We modified it.

Point 26: Line 270/Figure 9: Please revise the images, the mesh in the background is not different from the trace, please draw thicker or highlight in color, also the font is not readable

Response 26: We have revised the graph according to the recommendations and we made it more readable.

Point 27: Line 277/Figure 10: on the pictures in Fig. 10 it looks as if the clamping and pulling was not straight - if this is the case, all measured values of the tensile test are wrong - clamping must always be straight in all 3 axes

Response 27: If what you say is true you are correct. Unfortunately, the photo does not present the best angle and deforms the perspective and we decided to remove it, as not to give a false perspective. Thank you for your remark.

Point 28: Line 279 ff: up to line 360, the discussion reads like an introduction; in the discussion, your own values are to be classified in relation to known values from the literature. In addition, a critical consideration of the literature or one's own measured values can be made. In the present form, the discussion needs to be revised considerably. 

Reviewer 3 Report

Re: biomedicines-2604158

Comparison of the eggshell and the porcine pericardium membranes for guided tissue regeneration applications

Guided bone regeneration (GBR) is an important treatment method for dental implant therapy in the insufficient alveolar bone region for implant placement. A variety of bone graft materials and GBR membranes are now available, including absorbable and nonabsorbable materials. The safe and reliable materials have been developing.

This study is very interesting because it examines the possibility of using eggshell membranes as a GBR membrane.

There are some unclear points, which are listed below.

1. There was no clear statement why the eggshell membrane was focused and adopted for the GBR membrane.

2. There was a sentence “much research is needed to fully evaluate the properties and potential” in the Introduction section. The problems, advantages, and disadvantages detected in reference 16 and 17 should be described.

Since only the properties of eggshell were extracted in this study, it is not possible to determine whether or not the eggshell membrane can be used as a GBR membrane.

Author Response

Reviewer 3

Re: biomedicines-2604158

Comparison of the eggshell and the porcine pericardium membranes for guided tissue regeneration applications

Guided bone regeneration (GBR) is an important treatment method for dental implant therapy in the insufficient alveolar bone region for implant placement. A variety of bone graft materials and GBR membranes are now available, including absorbable and nonabsorbable materials. The safe and reliable materials have been developing.

This study is very interesting because it examines the possibility of using eggshell membranes as a GBR membrane.

There are some unclear points, which are listed below.

Point 1: 

  1. There was no clear statement why the eggshell membrane was focused and adopted for the GBR membrane.

Response 1: We have rephrased the paragraph and added the necessary details to further clear up the topic.

Point 2:

  1. There was a sentence “much research is needed to fully evaluate the properties and potential” in the Introduction section. The problems, advantages, and disadvantages detected in reference 16 and 17 should be described.

Response 2: We have added the details listed above. Thank you for your remark.

Point 3:

Since only the properties of eggshell were extracted in this study, it is not possible to determine whether or not the eggshell membrane can be used as a GBR membrane.

Response 3: Thank you for your remark. This is not a clinical study to determine the clinical efficiency of the membrane as a GBR material, nor an animal study, but it is a material study on the membrane to further understand the proprieties and components to see how they compare to other types of membranes which are currently in use.

Round 2

Reviewer 1 Report

The authors improved this manuscript in an organic way. I accept for the publication in present form. 

Author Response

Thank you very much for your effort and suggestions which have improved the quality of our manuscript!

Reviewer 2 Report

The changes to the manuscript are promising, some questions were answered, but unfortunately not fully implemented in the text. For example, the origin of various materials used is still missing: 1-butanol, ethyl alcohol. Remember this is a scientific publication (i.e. you have to be able to repeat it elsewhere).

 The authors used 2 different approaches for drying, this explains the different figures, in SEM (using vacuum) the two membranes are shown in the swollen state, while in AFM they are shrunk by drying in air, and therefore you see the fibers.

Line 277, 278 still comma instead of dot, furthermore the unit of the mentioned values is still missing [MPa].

 In addition, the discussion still reads like an introduction:

"... The structure relationship of the eggshell membrane has been extensively studied. The membrane behaves both as Mooney–Rivlin and Hookean materials in different environmental conditions. This means that it can stretch and restore its position, or it can have a nonlinear behavior such as rubber [35,36] ..."

This sentence could just as well be in the introduction - and this is just one example. 

The authors also could write it like this:

In their in vivo tissue response study, Radenković et al. [31] compared a commercially available cross-linked porcine sugar membrane with two non-cross-linked collagen membranes. They concluded that all membranes lead to similar bone formation, but the cross-linked membrane is more stable and resorbs more slowly up to 60 days after implantation.  "I would also write on right here instead of 

inserting a line break."

This reads much better instead of the "choppy" sentences, because of the line breaks after each sentence.

Author Response

The changes to the manuscript are promising, some questions were answered, but unfortunately not fully implemented in the text. For example, the origin of various materials used is still missing: 1-butanol, ethyl alcohol. Remember this is a scientific publication (i.e. you have to be able to repeat it elsewhere).

Response: Thank you. We modified the and added the necessary details for the substances used.

 The authors used 2 different approaches for drying, this explains the different figures, in SEM (using vacuum) the two membranes are shown in the swollen state, while in AFM they are shrunk by drying in air, and therefore you see the fibers.

Line 277, 278 still comma instead of dot, furthermore the unit of the mentioned values is still missing [MPa].

Response: Thank you. We modified the and added the necessary details.

 In addition, the discussion still reads like an introduction

"... The structure relationship of the eggshell membrane has been extensively studied. The membrane behaves both as Mooney–Rivlin and Hookean materials in different environmental conditions. This means that it can stretch and restore its position, or it can have a nonlinear behavior such as rubber [35,36] ..."

This sentence could just as well be in the introduction - and this is just one example. 

The authors also could write it like this:

In their in vivo tissue response study, Radenković et al. [31] compared a commercially available cross-linked porcine sugar membrane with two non-cross-linked collagen membranes. They concluded that all membranes lead to similar bone formation, but the cross-linked membrane is more stable and resorbs more slowly up to 60 days after implantation.  "I would also write on right here instead of 

inserting a line break."

This reads much better instead of the "choppy" sentences, because of the line breaks after each sentence.

Response: Thank you for your attention for details regarding the discussion section. We have rephrased the section for a better read and restructured the "choppy" sentences. We believe that the details that one might put in the introduction and very well of in the discussion section of our article due to the importance of understanding were our results fit in the bigger image of guided bone regeneration using a novel resorbable membrane. There are many factors to take into account and one might feel overwhelmed by all the details. Moreover there could be a much extensive comparison to similar research, but there is none to compare to. 

Reviewer 3 Report

Now this article achieves an acceptable level in the journal.

Author Response

(The authors gave the same response as above.)
